# Working Memory in Overweight Boys during Physical Education Classes

**DOI:** 10.3390/children10050805

**Published:** 2023-04-29

**Authors:** Aymen Hawani, Anis ben Chikha, Wael Zoghlami, Mohamed Abdelkader Souissi, Omar Trabelsi, Maher Mrayeh, Antonella Muscella

**Affiliations:** 1The Higher Institute of Sport and Physical Education (Ksar Saïd), University of Manouba, Manouba 2010, Tunisia; hawani.aymen@yahoo.com (A.H.); benchikhaanis@yahoo.fr (A.b.C.); mrayeh.meher@gmail.com (M.M.); 2Physical Activity, Sport and Health, Research Unit (UR18JS01), National Observatory of Sport, Tunis 1003, Tunisia; gaddoursouissi@yahoo.com (M.A.S.); trabelsi.omar@issepsf.u-sfax.tn (O.T.); 3High Institute of Sport and Physical Education, Sfax University, Sfax 3000, Tunisia; zoghlamiwael@gmail.com; 4Department of Biological and Environmental Science and Technologies (DiSTeBA), University of Salento, 73100 Lecce, Italy

**Keywords:** Sternberg Paradigm test, response time, ultimate frisbee, football, affective attitude, feeling scale

## Abstract

This study examined the effect of small-sided football games (SSFG) and small-sided Ultimate Frisbee games (SSUFG) on working memory, response times, and feeling mood of boys with overweight. Twenty-eight boys (age 12.23 ± 1.58 years) participated in two trials during physical education lessons (20-min SSFG and 20-min SSUFG) in a counterbalanced, randomized crossover design. The response times and feeling mood were measured for all participants post-exercise through the Sternberg paradigm and feeling scale. For the response times, the paired samples *t*-test revealed a significantly better reaction time after SSUFG at the “One item level” of the Sternberg paradigm (*p* = 0.014, Hedges’ *g* = 0.27, small effect) and (*p* = 0.010, Hedges’ *g* = 0.74, medium effect), at “Three item level” (*p* = 0.000106, Hedges’ *g* = 1.88, very large effect). The SSFG also showed vigor at the “Five item level” (*p* = 0.047, Hedges’ *g* = 0.61, medium effect). For the feeling mood, the feeling score was significantly higher after the SSUFG session than the SSFG session and the increase in feeling scores observed after switching from SSFG to SSUFG was significantly different (*p* < 0.001) from the decrease observed in feeling scores after switching from SSUFG to SSFG. Therefore, the results of the study allow the teacher to introduce new sports and reflect on the motor tasks he or she proposes for boys with overweight during physical education classes.

## 1. Introduction

During adolescence, physical activity levels tend to experience the most significant decrease [1]. According to van Sluijs et al. [2] physical activity levels drop considerably between the ages of 11 and 17, notably among adolescent females. Key findings of the study [2] showed that 81% of all 11–17-year-olds who were surveyed reported being physically inactive, with higher levels of physical inactivity reported among girls (84.7%) than boys (77.6%). Adolescents who increasingly engage in sedentary activities (e.g., watching TV or playing video games) during their daily routine are found to be less physically active and more involved in high-risk behaviors [3].

A long-standing body of evidence advocates for the regular practice of physical activity, as it entails a wide range of benefits for physical health, such as an increase in aerobic capacity, improvements in muscle strength, and more refined motor skills [4]. The benefits of physical activity are not limited to physical health, but also cover aspects of mental health, often leading to better academic performance [5]. 

It is therefore important to hone physical skills during childhood and early adolescence, not only because of their positive association with overall academic achievement, but also to improve body mass index and prevent overweight and obesity [6,7]. Zhang et al. [8] argue that children and adolescents should engage in at least 60 min of moderate- to high-intensity physical activity daily. Nevertheless, the effectiveness of structured physical activity opportunities among children has recently been questioned [9], since children tend to engage solely in activities that they find enjoyable [10]. Therefore, game-based physical activities can potentially offer a more attractive form of exercise to children [10]. One particular model of game-based physical activities is small-sided games [11]. This game model has the potential to offer better decision-making opportunities that can benefit children’s learning experience [11]. In addition, small-sided games can enhance integration, physical pleasure, physio-psychological responses, and mood states during PE sessions [12]. Indeed, the use of intense game-based activities, such as football, is an appealing modality given that children s’ usual activity patterns are high-intensity and intermittent in nature [13], as in team games [14]. 

Obesity and physical illiteracy, associated with low levels of physical activity, represent a real barrier for children overweight, involving the affective, behavioral, physical, and cognitive domains [15]. Structured interventions of evidence-based physical education programs could balance the potential negative effects of motor illiteracy and obesity during childhood [16]. Motivational activities in PE lessons have been found to improve the emotional states of children with obesity, which may contribute to their best advantage in effort intensity and physical enjoyment [17]. On the other hand, non-motivating running exercises negatively affect mood state and pleasure in children with obesity [18]. 

However, working memory (WM) plays an important role in children’s academic performance [17,18] and academic skills [19,20]. These associations have been also proven by the adverse influence of obesity on executive function, particularly WM [21,22].

Indeed, reduced memory, executive function, and increased impulsivity are some of the cognitive functions which appear as health consequences of obesity [23,24]. While other studies suggested that cognitive function in healthy children is not associated with high BMI [25,26]. 

While an intense bout of exercise has been shown to improve subsequent cognitive performance, even in adolescents, the effects of team games (of which soccer is the most popular and Ultimate Frisbee games) have received little or no attention. Taking into account the fact that the performance of children with obesity on WM tests is lower than that of children of normal weight [27,28], it is important to assess the effects of play-based exercise that children enjoy and chooses to participate, in a range of cognitive functions such as WM, particularly for those children with poor physical fitness, such as children with overweight or obesity [29,30].

Given that physical fitness is suggested as a key moderator of the exercise-cognition relationship [24,31], small-sided games-based exercises, such as Football or Ultimate Frisbee in physical education classes, can still be a valid mode of activity for children, given the ease of access to the equipment needed [32].

Thus, this research aimed to compare the effects of football small-sided games (SSFG) and Ultimate Frisbee small-sided games (SSUFG) on working memory, response times, and feeling mood of male children with overweight.

## 2. Materials and Methods

### 2.1. Participants

To detect a medium effect size *d* = 0.71 at an alpha of 0.05 with 95% power when conducting a paired sample t-test (the main statistical test to be used in this study), a priori calculation of the required sample size using the G*Power 3.1.9.4 software (Version 3.1, University of Dusseldorf, Dusseldorf, Germany) suggested that we would need 28 participants.

This study involved 28 overweight male children (M^age^ = 12.23 ± 1.58 years) who were enrolled in a public middle school in Tunisia. The boys’ body mass index (BMI) ranged from 25.0 to 30.0 (Table 1). 

The inclusion criteria for participants were as follows: (a) all participants must be enrolled in the same school class, (b) they must be attending two physical education classes per week (on Tuesdays and Fridays), and (c) they must have no prior competitive experience in football or ultimate frisbee. All participants were instructed to maintain their regular diet throughout the duration of the study and were required to report any injuries or illnesses incurred before or during the study.

Parents/guardians of the students received an informed consent letter containing detailed information about the study procedures. By signing the letter, they approved the participation of their dependents. The study was conducted according to the guidelines of the 2013 Helsinki Declaration. The experimental protocol was approved to comply with the guidelines of the 2013 Helsinki Declaration by the local Research Ethics Committee at the High Institute of Sports and Physical Education of Ksar- Saïd (Approval reference: N°02/2021).

### 2.2. Study Design 

This study was conducted during the 2021–2022 school year over the course of four weeks. Two experimental sessions were carried out on an artificial grass pitch, between 9:00 and 10:00 a.m., while the temperature ranged from 19 to 20 °C, and the wind fluctuated between 6 to 8 km·h^−1^. 

During the first week (familiarization phase), participants were familiarized with the intervention and the testing procedures. The boys were introduced and familiarized with the fundamental technical-tactical aspects and rules of the SSUFG as well as the SSFG. A full understanding of the intervention and testing procedures was constantly verified at this stage. 

In the second week, the boys were randomly divided into two groups of 14 each, which allowed the implementation of a counterbalancing procedure. 

In the third and fourth weeks (experimental phase), each of the two groups took part in a 20-min SSUFG and a 20-min SSFG in a counterbalanced crossover design. Indeed, the order of sessions was randomly counterbalanced, ensuring that half of the participants completed the SSUFG before the SSFG, while the other half competed for the intervention in a reversed order (Table 2). 

### 2.3. Practical Intervention 

During two physical education classes (one on the third and another on the fourth week), each of the two groups (*n* = 14) played either a football or an ultimate frisbee small-sided game (7 vs. 7) on a minimized artificial grass pitch (20 m × 40 m). The lineup of each team was randomly set up. The small-sided games were preceded by a standardized 15-min warm-up, including 5 min of running, 5 min of dynamic stretching, and 5 min of the ball (in case of an SSFG) or disc drills (in case of an SSUFG).

The SSFG and SSUFG consisted of two 10-min play periods interspersed by a 5-min recovery period. Players had to follow the offensive and defensive rules of football or ultimate frisbee, depending on the task at hand. 

### 2.4. Data Collection

#### 2.4.1. Anthropometric Data

The BMI was calculated as the body mass (in kilograms) divided by the square of the height (in meters) (kg/m^2^). With subjects barely clad, body mass was measured with precision up to 0.1 kg using a portable digital beam scale (Tanita, Tokyo, Japan). Body height was measured to the nearest millimeter with children standing upright against a Holstein portable stadiometer, while barefoot or wearing thin stockings (Tanita).

BMI values were used to define the obesity indices (overweight, obesity, severe obesity) according to the International Obesity Task Force (IOTF) cut-offs [33]. 

#### 2.4.2. Cognitive Function Test

The cognitive function test, known as the Sternberg Paradigm test [34], was carried out on a Lenovo ThinkPad T470s laptop and took approximately 8 min to complete. The test instructions were displayed on the screen before the start of each level of the test. Students were allowed a rehearsal time before the start of each level, immediately after the presentation of the instructions. No data were collected during rehearsal. Participants completed the test individually in a quiet and isolated environment, avoiding any form of interaction. They wore noise-canceling headphones, and the room was dimly lit to reduce external distractions and improve the visibility of the screen.

The Sternberg paradigm is a test that assesses working memory and has three different levels, with each level involving a distinct working memory load of either one, three, or five items. In the “one-item” level of the test, the target was consistently the number “3” and there were 16 stimuli presented. In the “three-item” and “five-item” levels, the target was a set of three or five letters, respectively, that were randomly generated. Each of the “three-item” and “five-item” levels had 32 stimuli presented. At the start of each level, the target items were displayed on the screen, along with clear instructions to press the right arrow key if the item was a target and the left arrow key if it was a distractor. For each level of the test, the correct response was distributed equally between the left and right arrow keys and choice stimuli were consistently displayed in the middle of the screen with a 1-s interval between stimuli. The focus of the study was on the response time (in milliseconds) for target items, once correctly identified, along with the accuracy of response (percentage % of correct responses).

#### 2.4.3. The affective Component of Exercise

The affective component of exercise was measured using the Feeling Scale (FS), developed by Hardy and Rejeski [35,36] to assess an individual’s affective response to physical activity. The 11-point scale consists of a series of numbers ranging from −5 to +5, with each number anchored by verbal descriptions of affective states.

The participants responded to the question, “How do you feel right now?” [37]. The anchors of the scale are as follows: −5 = Very bad, −4 = Bad, −3 = Fairly bad, −2 = Somewhat bad, −1 = Slightly bad, 0 = Neutral, +1 = Slightly good, +2 = Somewhat good, +3 = Fairly good, +4 = Good, +5 = Very good. Participants were instructed to rate their affective state 5 min after each SSFG or SSUFG by selecting the number that corresponded to the anchor that best represented their feelings at that moment [38].

### 2.5. Statistical Analysis

Data were expressed as mean ± standard deviations (SD) in the text, tables, and figures. The normality of data was assessed using the Shapiro-Wilk test. In the case of non-normal distribution, nonparametric tests were used. Data collected using the Sternberg Paradigm test were analyzed using the paired samples *t*-test. The significance level was set at *p* < 0.05. The Hedges’ *g* coefficient was calculated to determine the magnitude of differences between data collected following SSFG and SSUFG.

For the feeling scale, analyses were conducted using the Mann-Whitney U test to compare changes in feeling scale scores (Δ) recorded in group 1 and those recorded in group 2. Effect sizes were reported as Pearson’s *r*. The significance level was set at *p* < 0.05. To eliminate extremely fast and slow responses (deemed unreasonable), the response time was analyzed using minimum (<200 ms) and maximum (1500–3000 ms, depending on task difficulty) thresholds.

## 3. Results

### 3.1. Cognitive Function

In group 1, the statistical analyses revealed a significant difference in the best response time recorded at the “One-item” level after the SSUFG (*p* < 0.05, Hedges’ *g* = 0.27, small effect) compared to that recorded after the SSFG. On the other hand, no significant differences were observed at the “Three-item” (*p* > 0.05) and “Five-item” levels (*p* > 0.05, Table 3).

In group 2, greater scores of response time were recorded following SSUFG at the “One-item” level (*p* < 0.01, Hedges’ g = 0.74, medium effect) and the “Three-item” level (*p* < 0.001, Hedges’ g = 1.88, very large effect). On the other hand, a significant difference in favor of the SSFG was reported at the “Five-item” (*p* < 0.05, Hedges’ g = 0.61, medium effect Table 4).

### 3.2. Affective Component of Exercise

When comparing the scores of the feeling scale collected from groups 1 and 2, the statistical analyses revealed that there was a significant increase in scores when switching from SSFG to SSUFG (*p* < 0.001). Conversely, a significant decrease in scores when the students switched from SSUFG to SSFG (group 2, Table 5 and Figure 1). Specifically, participants reported more positive affective responses to the SSUFG compared to the SSFG.

## 4. Discussion

This study compared the effects of football small-sided games (SSFG) and Ultimate Frisbee small-sided games (SSUFG) on response time and the affective response of overweight boys. The main findings revealed that the SSUFG had a significantly positive impact on both response time and affective response in overweight boys. Specifically, response time was shorter, particularly at the “One-item” level in group 1, and at the “One-item” and “Three-item” levels in group 2 after the SSUFG session. More interestingly, the SSUFG session resulted in a significantly more positive affective response in the overweight boys compared to the SSFG session.

The use of SSUFG in simulated competitive settings provides a promising framework for improving cognitive functions in school-aged students [39]. This is because ultimate frisbee involves high technical and tactical demands, which can challenge and stimulate cognitive processes such as decision-making, problem-solving, and spatial awareness [39]. Moreover, by creating a competitive environment, SSUFG can help students develop important non-cognitive skills such as teamwork, communication, and resilience [39]. While physical fitness is important for optimal performance in ultimate frisbee, the present findings suggest that even overweight boys can benefit from participating in this activity.

The beneficial effects of SSUFG on working memory highlighted in the present study may be explained by the social interactions that abound during this activity. Research has shown that social interaction can enhance cognitive performance, possibly through the activation of brain regions involved in reward processing and social cognition [40]. Undoubtedly, games-based activities require a significant degree of social interaction and cognitive stimulation, which are crucial determinants of neuroplasticity [41]. This latter refers to the brain’s remarkable ability to adapt to changes in the environment through the creation of new neural connections and the strengthening of existing ones. This may also explain the beneficial effects of SSUFG on working memory [41].

Ultimate frisbee is a sport that places great emphasis on communication and upholding the values that are integral to the spirit of the game [42]. Through the cultivation of these values, a distinct culture can emerge in an ultimate frisbee team, strengthening the social dynamics between the players [42] In recent studies, young ultimate frisbee players showed a lower level of neuroticism [43,44], a higher level of extraversion and they were characterized by average personality indicators [45]. Research has shown that the enjoyment of play and personality indicators (behaviors/well-being) influence cognitive performance [46].

When examining the affective responses to SSUFG and SSFG, it was found that both models of small-sided games resulted in a positive affective response to physical education classes. This is in line with previous studies that had reported positive affective responses to small-sided games [12,38].

Following the SSUFG session, the boys reported a significantly more positive affective response compared to the SSFG session. This difference in affective response could potentially be attributed to the fact that the overweight boys were more actively involved and committed to participating in the SSUFG. The findings of this study suggest that SSUFG may increase the likelihood of experiencing positive feelings during physical activity. This is particularly significant for overweight children, as positive feelings can help to sustain exercise and ultimately lead to health and fitness benefits [32]. Moreover, enjoyment during the SSUFG seems to have a mediating effect between autonomy support and performance [47]. Previous research has also provided evidence to support the notion that SSUFGs contribute to sustained positive feelings due to their unique social features. These features include self-refereeing, collective arbitration, self-regulation, and independent communication, which set it apart as a distinct team sport [46]. Ultimate frisbee players have personality profiles similar to players in other team sports [45]. The distinctive forms of cooperation existing among players in ultimate frisbee are the result of the norms enforced during competition, based on a combination of self-discipline and concerns about the reputation of players and teams [47]. Ultimate Frisbee takes on the characteristics of a moral community organized around the spirit of the game [47]. Young people are encouraged to strive for personal excellence and competitive success at the same time, valuing fairness, and respect for both the rules and their opponents [48].

As social interactions play a critical role in determining the level of enjoyment in team sports [28], the SSUFG session could potentially have made the activity an appealing exercise model for overweight boys. On top of that, the boost in positive affective responses to the SSUFG sessions during physical education classes can serve as a significant indicator of the positive environment created among overweight boys [49].

Lastly, this study suggests that the body image of the boys who took part was not a hindrance to their adherence to the SSUFG sessions. The findings of the present study contrast with previous research conducted by Foley Davelaar [50], which suggested that an increased body mass index can discourage physical activity among children between the ages of 6 and 11. That said, after the age of 12, body image becomes a more significant factor in determining the level of engagement in physical activity.

The present study has certain limitations that should be considered when interpreting the results. Firstly, the study only included male children, which limits the generalizability of the findings to female children. Therefore, future research should consider recruiting female children to compare the impact of ultimate frisbee small-sided games on working memory between the two genders. Secondly, it is important to note that exercise and physical fitness are crucial factors in the mechanisms underlying the effects of physical activity on cognitive and affective responses. Hence, future research should consider including physiological measures, such as heart rate and oxygen consumption. Such measures would provide valuable information on the mechanisms by which ultimate frisbee small-sided games influence cognitive and affective outcomes in overweight children. Thirdly, this study did not evaluate the selection of organizational and communicative methods used during small-sided games in physical education classes, nor did it investigate differences in productive (indirect) and reproductive (direct) teaching styles. Future research should consider examining these factors, as they may impact the effectiveness of small-sided games in improving cognitive and affective outcomes in overweight children.

An area that could be explored in future research is the impact of SSUFG and SSFG on both overweight and non-obese children, with a larger sample size for both genders. It may also be worthwhile to consider other variables such as geographic location (urban or rural), age (primary or secondary school), as well as levels of fatigue, vigor, and confusion during physical education classes.

## 5. Conclusions and Practical Implications

This study builds on prior research and confirms that small-sided games can enhance working memory and affective response in young boys. The key findings revealed that playing an ultimate frisbee small-sided game for 20 min resulted in a shorter response time and a more positive affective response among boys with overweight compared to a football small-sided game. These findings provide valuable insights into the potential benefits of small-sided games and highlight the importance of considering the specific type of game played when designing physical education lessons adapted to children with overweight. Although physical demands during ultimate frisbee games may be higher for boys with overweight, the positive impact of this activity on cognitive and non-cognitive functions could have helped offset this challenge. Therefore, the use of ultimate frisbee small-sided games can be an effective strategy for promoting physical activity and cognitive development among students, regardless of their weight status.

## Figures and Tables

**Figure 1 children-10-00805-f001:**
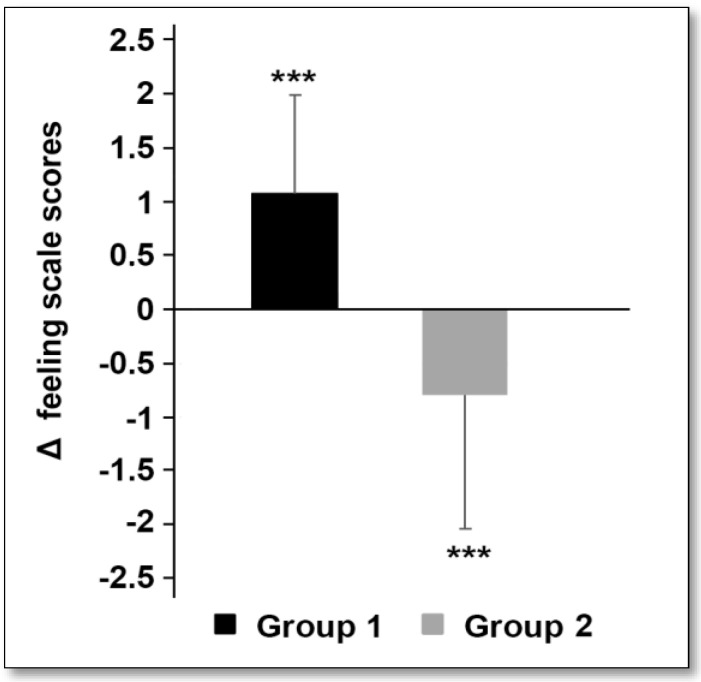
Δ Feeling scale scores in the two groups. The results are presented as mean ± SD; *** *p* < 0.001.

**Table 1 children-10-00805-t001:** Anthropometric characteristics of participants (Mean ± SD).

Participants (*n* = 28)	
Height (cm)	152 ± 0.84
Weight (kg)	62.13 ± 0.21
BMI (kg/m^2^)	26.89 ± 0.15

SD, Standard Deviation; BMI, Body Mass Index.

**Table 2 children-10-00805-t002:** Description of the experimental protocol and the counterbalancing procedure.

		Experimental Period				
		**Testing sessions**				
	First week	Second week	Third week		Fourth week	
	Familiarization sessions	Resting trial	Session 1		Session 2	
			*SSFG*	*SSUFG*	*SSFG*	*SSUFG*
Group 1 (*n* = 14)			X			X
Group 2 (*n* = 14)				X	X	

X = performed activity.

**Table 3 children-10-00805-t003:** Data collected using the Sternberg Paradigm test for group 1.

**Group 1**	**Sternberg Paradigm**	**Δ% SSFG**	**Δ% SSUFG**	**T**	** *p* **	**Hedges’ *g***
	“One-item” level	−3.82± 7.36%	−5.75 ± 6.67%	2.826	0.014 *	0.27 (Small effect)
	“Three-item” level	−4.09 ± 3.31%	−3.71 ± 4.68%	−0.393	0.701	***
	“Five-item” level	+0.96 ± 5.12%	−0.81 ± 3.91%	1.322	0.209	***

SSFG, football small-sided game; SSUFG, ultimate frisbee small-sided game; *, *p* < 0.05; ***, *p* < 0.001

**Table 4 children-10-00805-t004:** Data collected using the Sternberg Paradigm test for group 2.

**Group 2**	**Sternberg Paradigm**	**Δ% SSFG**	**Δ% SSUFG**	**T**	** *p* **	**Hedges’ *g***
	One item	−5.67 ± 4.69%	−1.87 ± 5.52%	−3.006	0.010 *	0.74 (Medium effect)
	Three item	−6.54 ± 2.90%	+4.58 ± 7.83%	−5.480	0.000106 *	1.88 (Very large effect)
	Five item	+0.63 ± 4.32%	−1.82 ± 3.67%	2.190	0.047 *	0.61 (Medium effect)

SSFG, football small-sided game; SSUFG, ultimate frisbee small-sided game; *, significant difference.

**Table 5 children-10-00805-t005:** Feeling scale scores were collected from groups 1 and 2.

	Test 2–Test 1 (Δ)	Mann-Whitney U Test				
Group 1	(SSUFG–SSFG) = 1.07 ± 0.92	Mean rank = 19.93	U = 22.00	Z = −3.59	*p* < 0.001 *	*r* = 0.48 (Medium effect)
Group 2	(SSFG–SSUFG) = −0.79 ± 1.25	Mean rank = 9.07				

SSFG, football small-sided game; SSUFG, ultimate frisbee small-sided game; *, significant difference.

## Data Availability

Not applicable.

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
