# Peer review of "Working Memory in Overweight Boys during Physical Education Classes"

_children, 2023, doi:10.3390/children10050805_

Round 1
Reviewer 1 Report
1. Please check the format of the draft. There are some mistakes in the form.
2. The participants of the study are only boys that also be written in the discussion. However, why in the title, the abstract and the conclusion be changed to children or boys and girls? How can the results of the boys be inferred to the boys and girls?
Author Response
Dear Reviewer
We appreciate the time and effort that you dedicated to providing feedback on our manuscript and we feel grateful for the insightful comments and valuable suggestions. All changes are highlighted in yellow throughout the revised manuscript.
Please see below a point-by-point response to your comments
Comment 1 Please check the format of the draft. There are some mistakes in the form.
Response The reviewer is correct. Format and style have now been throughout verified.
Comment 2 The participants of the study are only boys that also be written in the discussion. However, why in the title, the abstract and the conclusion be changed to children or boys and girls? How can the results of the boys be inferred to the boys and girls?
Response We agree with the reviewer. We, therefore, changed “children” or “adolescents” to “boys” throughout the manuscript.
Reviewer 2 Report
Dear authors:
Thanks for submitting your article on this interesting topic.
The effect of PA on cognitive functions and working memory is an interesting topic.
Nevertheless, the manuscript has some flaws which will be detailed here:
Abstract: some indications should be added in the abstract that the trials were conducted during PE classes.
Introduction: this section appears as insufficient. Please add more recent works to your introduction section.
There is a continuous confusion using “children”, “youth people” (young people???) and “adolescents” in the manuscript. Please be consistent in this sense.
Please improve also the structure of phrases.
The fact that only 28 school students participated is a major flaw of the study. Even if you try to justify this by your sample analysis. Why didn’t you include girls into your sample?
How was BMI measured?
What about the possible effect of former PA in overweight adolescents? Did you take this into account? If not, why not?
Discussion:
There is no need to repeat the results of the tables in the discussion section.
Again, confusion between children and adolescents.
Limitations:
I would prefer to see a specific limitation section. As, due to the small sample no generalization is possible, and no girls where included, it diminishes the scientific value of the paper.
Conclusion:
The practical implications of using PA, and specifically SSG in PE, are unclear and insufficient.
Author Response
Dear
We appreciate the time and effort that you dedicated to providing feedback on our manuscript and we feel grateful for the insightful comments and valuable suggestions.
Please see below a point-by-point response to your comments
Comment 1 Abstract: some indications should be added in the abstract that the trials were conducted during PE classes.
Response We agree with the reviewer’s point of view. The study context has been clearly stated in the abstract.
Comment 2 Introduction: this section appears as insufficient. Please add more recent works to your introduction section.
Response The reviewer is correct. The “Introduction” section has been enriched with the further theoretical grounding and more recent references.
Indeed, it was added:
During adolescence, physical activity levels tend to experience the most significant decrease [1]. According to van Sluijs et al. [2] physical activity levels drop considerably between the ages of 11 and 17, notably among adolescent females. Key findings of the study [2] showed that 81% of all 11-17-year-olds who were surveyed reported being physically inactive, with higher levels of physical inactivity reported among girls (84.7%) than boys (77.6%). Adolescents who increasingly engage in sedentary activities (e.g., watching TV or playing video games) during their daily routine are found to be less physically active and more involved in high-risk behaviors [3].
A long-standing body of evidence advocates for regular physical activity, as it entails a wide range of benefits for physical health, such as an increase in aerobic capacity, improvements in muscle strength, and more refined motor skills [4]. The benefits of physical activity are not limited to physical health, but also cover aspects of mental health, often leading to better academic performance [5].
- Eime, R.M.; Young, J.A.; Harvey, J.T.; Charity, MJ.; Payne, W.R. A systematic review of the psychological and social benefits of participation in sport for children and adolescents: informing the development of a conceptual model of health through sport. Int J Behav Nut Phys Act.2013 ;10:98–119.
- van Sluijs E.M.F.; Ekelund U.; Crochemore-Silva I.; Guthold R., Ha. A.; Lubans D.; Oyeyemi A.L.; Ding D.; Katzmarzyk PT. Physical activity behaviours in adolescence: current evidence and opportunities for intervention. Lancet. 2021 Jul 31;398(10298):429-442. doi: 10.1016/S0140-6736(21)01259-9. PMID: 34302767; PMCID: PMC7612669.
- Verónica Cabanas-Sánchez; David Martínez-Gómez; Irene Esteban-Cornejo; Alejandro Pérez-Bey; José Castro Piñero & Oscar L. Veiga. Associations of total sedentary time, screen time and non-screen sedentary time with adiposity and physical fitness in youth: the mediating effect of physical activity. Journal of Sports Sciences, 2019. DOI: 1080/02640414.2018.1530058
- Gabriela P. Teixeira; Kisian C. Guimarães; Ana Gabriela N.S. Soares; Elaine C. Marqueze; Cláudia R.C. Moreno; Maria C. Mota, Cibele A Crispim. Role of chronotype in dietary intake, meal timing, and obesity: a systematic review, Nutrition Reviews, Volume 81, Issue 1, January 2023, Pages 75–90, https://doi.org/10.1093/nutrit/nuac044
- Rodriguez-Ayllon M.; Cadenas-Sánchez C.; Estévez-López F.; Muñoz N.E.; Mora-Gonzalez J.; Migueles J.H.; Molina-García P.; Henriksson H.; Mena-Molina A.; Martínez-Vizcaíno V.; Catena A.; Löf M.; Erickson K.I.; Lubans D.R.; Ortega F.B.; Esteban-Cornejo I. Role of Physical Activity and Sedentary Behavior in the Mental Health of Preschoolers, Children and Adolescents: A Systematic Review and Meta-Analysis. Sports Med. 2019. PMID: 30993594.
Comment 3 There is a continuous confusion using “children”, “youth people” (young people???) and “adolescents” in the manuscript. Please be consistent in this sense.
Please improve also the structure of phrases.
Response We agree with the reviewer. We, therefore, changed “children” or “adolescents” to “boys” throughout the manuscript. Format and style have now been throughout verified.
Comment 4 The fact that only 28 school students participated is a major flaw of the study. Even if you try to justify this by your sample analysis. Why didn’t you include girls into your sample?
Response In this study, a pediatrician classified all participants as prepubertal (stage 1) according to Tanner's criteria (Tanner, 1981). Regrettably, 15 participants (4 male and 11 female) were excluded before the start of this experiment because they were adolescents. In addition, previous studies have shown that menstrual cycle phases affect physical performance and decision-making (Julian et al., 2017; Croteau, 2015). For this reason, female participants were excluded from the study.
1. Tanner, J. M. (1981). Growth and maturation during adolescence. Nutrition Reviews, 39(2), 43–55.
2. Julian R, Hecksteden A, Fullagar HH, Meyer T. The effects of menstrual cycle phase on physical performance in female soccer players. PLoS One. 2017 Mar 13;12(3):e0173951. doi: 10.1371/journal.pone.0
3. Croteau, C. M. (2015). Menstruation & physical activity: A test of the theory of planned behavior and menstrual
Comment 5 How was BMI measured?
Response We agree with the reviewer that there were some details concerning the BMI measurements missing in this section. We now, in the “Participant” subsection, added:
“Body mass index (BMI) was computed by dividing body mass (kg) by height (m) squared (kg/m²).”
And in the Anthropometric Parameters subsection we added:
“For children and adolescents, the Center for Disease Control and Prevention defines overweight as a body mass index (BMI: weight in kilograms divided by height in meters squared) between the 85th and 95th percentiles and obesity as a BMI at or above the 95th percentile for sex and age (Kuczmarski et al., 2002)”.
Comment 6 What about the possible effect of former PA in overweight adolescents? Did you take this into account? If not, why not?
Response The sample recruited in this study is characterized by low physical activity levels and has practically the same physical education experience. In the inclusion criteria, we mentioned that they had no previous competitive experience in Football and Ultimate Frisbee.
Comment 7 Discussion: There is no need to repeat the results of the tables in the discussion section.
Again, confusion between children and adolescents.
Response We agree with the reviewer. We, therefore, changed “children” or “adolescents” to “boys” throughout the manuscript.
Comment 8 I would prefer to see a specific limitation section. As, due to the small sample no generalization is possible, and no girls were included, which diminishes the scientific value of the paper.
Response The limitation of the study is now discussed in the revised manuscript. We thank the reviewer for their suggestion.
Comment 9 The practical implications of using PA, and specifically SSG in PE, are unclear and insufficient.
Response Practical implications are now available in the revised manuscript, precisely in the “Conclusion” section, we added:
“The results showed that playing Small-Sided Ultimate Frisbee for 20 minutes resulted in faster response times and a positive mood state. These findings provide new evidence that Small-Sided Games have advantageous effects on working memory and post-exercise mood in overweight boys. Thus, based on the study results, it is recommended that physical education teachers incorporate new sports and consider the type of motor tasks assigned to boys with overweight. These findings have practical applications for both physical education teachers and coaches, as they emphasize the importance of Small-Sided Games in promoting a reduction in sedentary behavior among boys with overweight.”
Reviewer 3 Report
Authors have made a great effort to perform a good study. They have partly met their goal as specific statistical flows have to be corrected:
Major comments
a. Instead of paired t-test they should contact its non-parametric equivalent: Wilcoxon signed rank test. The reason is mainly the small sample each time which is 14 subjects. Alternatively they could profit from ssev.eu.
b. They provide the ANOVA models effect size measure η2 whithout explaining why. They also calculate Cohen d. Nevertheless, they should definitely calculate the Hedges's g correction of bias of Cohen d for small samples in both cases. Alternatively they could profit from ssev.eu. A forest plot would also provide more information any way.
c. In abstract and Table 3 we see that the 5-item reaction time had limit difference (0.047) and NOT significant. This means that part of the 14 subjects group-2 had significant difference and others did not. The statistical method has to be changed and if the limited result persists, it has to be discussed accordingly (and explained). The characteristics of this group might help them design a more focused study in the future.
Minor comments:
Syntax, grammar and typing errors have to be corrected all along the text.
Author Response
Dear
We appreciate the time and effort that you dedicated to providing feedback on our manuscript and we feel grateful for the insightful comments and valuable suggestions.
Please see below a point-by-point response to your comments
Comment 1 Instead of paired t-test they should contact its non-parametric equivalent: Wilcoxon signed rank test. The reason is mainly the small sample each time which is 14 subjects. Alternatively they could profit from ssev.eu.
Response We understand the reviewer’s concerns. That said, it is widely established that using non-parametric tests requires rejecting the normality assumption. In our case, all datasets collected on “reaction time” were confirmed to be normally distributed using the Shapiro-Wilk test, thereby allowing the use of the parametric test for pairwise comparisons of two dependent groups (i.e., t-test for paired samples). On the other hand, the non-parametric alternative (i.e., Mann Whitney U Test) was performed on data collected using the Feeling Scale, as some of those datasets failed to meet either the assumption of normality or homogeneity of variances.
The web-based application of SSEv is a useful tool for small-sample statistical analysis. However, to the best of our knowledge, its efficiency is not comparable to the giant of statistics “SPSS”.
Comment 2 They provide the ANOVA models effect size measure η2 whithout explaining why. They also calculate Cohen d. Nevertheless, they should definitely calculate the Hedges's g correction of bias of Cohen d for small samples in both cases. Alternatively they could profit from ssev.eu. A forest plot would also provide more information any way.
Response We apologize for the omission. Effect sizes for the MWU test should be reported as Pearson correlation coefficient (r) or U-statistic. It was only a typo, and it has now been corrected. Furthermore, Hedges’ g was calculated and reported instead of Cohen’s d, knowing that both values are widely used as estimates of effect size for t-tests, even when small samples are involved.
Comment 3 In abstract and Table 3 we see that the 5-item reaction time had limit difference (0.047) and NOT significant. This means that part of the 14 subjects group-2 had significant difference and others did not. The statistical method has to be changed and if the limited result persists, it has to be discussed accordingly (and explained). The characteristics of this group might help them design a more focused study in the future.
Response We thank the reviewer for their comment. We would like to explain that the comparison, in this case, was a within-subject one, only the order of treatment was reversed in the second measurement; meaning that there’s no 2nd group. We agree with the reviewer that the significance level was poor (however, still the difference was significant at 0.05). Here the effect size intervenes to “save the night”, as it indicates there’s a medium effect, a respected effect.
We kindly ask the reviewer to state if a re-discussion of the results is still needed after reading the above response to their comment.
Comment 4 Syntax, grammar and typing errors have to be corrected all along the text.
Response The manuscript has been proofread for syntax and grammatical errors.
Round 2
Reviewer 1 Report
Good work. Please check the draft following the template.
Author Response
Thank you.
Done!
Reviewer 2 Report
Dear authors. Thanks for answering correctly all my questions.
Author Response
Thank you for contributing to the enhancement of the quality of our manuscript.
Author Response

(The authors gave the same response as above.)
